# Design and Realization of a Compact Efficient Beam Combiner, Based on Liquid Crystal Pancharatnam–Berry Phase Gratings

**Boxuan Gao** [ID], **Jeroen Beeckman** [ID] and **Kristiaan Neyts** *[ID]

Electronics and Information Systems Department, Ghent University, 9052 Ghent, Belgium;
boxuan.gao@ugent.be (B.G.); jeroen.beeckman@ugent.be (J.B.)
* Correspondence: kristiaan.neyts@ugent.be; Tel.: +32-9-264-3381

**Abstract:** We demonstrate a laser beam combiner based on four photo-patterned Pancharatnam–Berry (PB) phase gratings, which is compact and has high diffraction efficiency for incident circularly polarized light. The nematic liquid crystal mixture E7 is used as anisotropic material, and the thickness of the layer is controlled by spacers. The beam combiner can bring two parallel laser beams closer to each other while remaining parallel. This work shows the potential to realize components based on flat optical LC devices.

**Keywords:** geometric phase grating; diffraction; laser beam combiner; etendue reduction





## 1. Introduction

In conventional lenses, the optical path difference (OPD) is modulated because the thickness of the glass varies with position. Fresnel lenses can yield a similar functionality with thin layers, but the discontinuous structures lead to unwanted scattering losses, and the resolution is often limited. To tackle these problems, optical elements have been developed based on the Pancharatnam–Berry (PB) phase. In such devices, the phase of the wavefront is modulated by going through a continuous variation of the polarization state [1]. This method has been studied for applications in beam steering [2], beam shaping [3], waveguides [4], augmented reality (AR), and virtual reality (VR) [5,6]. PB phase gratings based on the photoalignment of liquid crystal (LC) as anisotropic material can achieve diffraction efficiencies of more than 99% [7]. When the thickness of the anisotropic layer corresponds to a half-wave plate, and the azimuth angle of the director increases linearly with the lateral position, then incident right-handed circularly polarized light is efficiently diffracted into the first diffraction order, becoming left-handed circularly polarized light. This is because a half wave plate inverses the polarization of light on the Poincaré sphere (right-handed to left-handed polarization) and the PB grating adds a position-dependent phase shift [3].

One method to fabricate the PB phase grating is based on surface-assisted photoalignment [8–10]. When an azo-dye photoalignment material is exposed to linearly polarized UV or blue light, a fraction of molecules will tend to reorient with their long axis perpendicular to the polarization direction [11]. The director of the liquid crystal that is deposited onto the alignment layer will follow the same direction. This method can be used for a rather wide working wavelength range [2,12,13]. In a geometric phase grating or Pancharatnam–Berry phase grating, the orientation of the slow axis varies continuously. Such a pattern can be realized by exposing the photoalignment material to the interference of two coherent blue or UV laser beams with orthogonal circular polarization that are incident under an angle. The continuous variation avoids abrupt phase resets [14], and results in high diffraction efficiency. Other fabrication methods, such as direct writing, allow more freedom in the photoalignment pattern, and an arbitrary wavefront is achievable [9].

Diode lasers have obtained excellent reliability and power and cost efficiency. They are increasingly used in applications that require high-power laser light, such as solid-state

laser pumping [15], industrial processing, and laser projection systems [16]. Often, it is desired to couple the laser light of all diodes into an optical fiber. It is a challenge to increase the beam power of an individual diode laser above a few Watts, and therefore, arrays of eight or more diode lasers, called diode laser banks, are used to reach higher total power. The axes of the individual laser beams generated by a diode laser bank are parallel, but in order to provide cooling and mechanical stability, the laser diodes are separated by a distance on the order of a cm. For many applications, it is interesting to reduce the lateral distance between the parallel laser beams to few mm, while keeping the laser beams parallel. As we want the axes of the beams to be parallel in the end, the laser beams should be refracted at least twice. In this way, the average laser power per unit area (in W/mm$^2$) can be increased considerably. The spacing can be reduced by using refractive components to change the propagation direction of the laser beams [17], such as the lens duct technique [15], the gradient index rod [18], stripe mirror plates, and V-stack mirrors [19]. Such systems rely on bulky optical components and result in heavy optical systems. Moreover, the alignment of many refractive components on the scale of a few mm can be a challenge.

In this work, we propose a laser beam combiner by geometric phase gratings. The first component uses a diffraction grating for each laser beam to change the propagation direction of the beams and bring the beams closer together. The second component uses two diffraction gratings to make the beams parallel again, but now at a closer lateral distance. To demonstrate the principle, we have implemented a compact and efficient laser combiner for two circularly polarized red laser beams. This design is based on two liquid crystal samples, realized by photoalignment. Each sample contains two diffraction gratings next to each other, with opposite LC rotating directions, as shown in Figure 1.

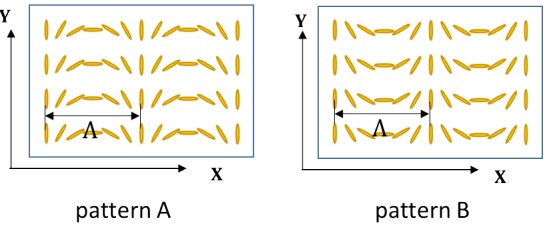

**Figure 1.** Photoalignment patterns A and B to diffract circularly polarized light efficiently. Pattern A diffracts left-handed circularly polarized incident light towards the right.

## 2. Materials and Methods

### 2.1. Liquid Crystal Sample Preparation

For the realization of the diffractive components, we use unpolished float glass substrates coated with indium tin oxide (ITO). The substrates are cleaned (in soap, acetone, isopropanol, and deionized water) and dried in an oven at 100 °C. Before spin casting, the substrates are treated by UV-Ozone for 20 min. The photoalignment material Brilliant Yellow (BY) is diluted in Dimethylformamide (DMF) at 0.5wt%, and the solution is filtered and spin-coated on the ITO side of the substrates (3000 rpm for 30 s). The coated substrates are then baked on a hotplate at 90 °C for 10 min, to evaporate the solvent residue.

The LC cell is realized by gluing two substrates together, using a UV curable liquid photopolymer (Norland Optical Adhesive 68) mixed with spacers. The liquid crystal material used here is E7, with a clearing temperature of 59 °C and a relatively high birefringence, allowing to obtain high diffraction efficiency even under a large diffraction angle [20]. E7 has wavelength-dependent refractive indices [21], with the ordinary refractive index $n_o = 1.52$ and birefringence $\Delta n = 0.21$ at 633 nm wavelength. The diffraction in the first and zero orders depends on the wavelength according to Equations (1) and (2) [22], where $\eta_0$ and $\eta_{\pm 1}$ denote the diffraction efficiency for the zero and ($\pm$) first order, respectively; $d$

is the LC layer thickness, $\lambda$ is the incident wavelength, and $S'_3 = S_3/S_0$ is the normalized Stokes parameter that describes the ellipticity of the incident light.

$$\eta_{\pm 1} = \frac{1}{2}(1 \mp S'_3)sin^2(\frac{\pi \Delta nd}{\lambda}) \tag{1}$$

$$\eta_0 = cos^2(\frac{\pi \Delta nd}{\lambda}) \tag{2}$$

These formulas are for pattern A. The efficiency for the first diffraction order (+1, diffraction towards the right) for left-handed circularly polarized incident light ($S'_3 = -1$) with wavelength 633 nm reaches a maximum of 100% at the optimal thickness of 1.51 μm.

The thickness of the empty cell is estimated from the transmission spectrum, obtained with a Perkin Elmer Lambda 35 UV/VIS spectrophotometer. Typically, the thickness deviates somewhat from the desired value, resulting in the optimal diffraction for a different wavelength, and the reduction of the diffraction efficiency at the target wavelength (633 nm).

### 2.2. Photoalignment Patterning and Fabrication

The procedure to fabricate the LC phase grating is illustrated in Figure 2. The empty BY-coated LC cell is exposed to the interference pattern of two orthogonal, circularly polarized ultraviolet (UV) beams, as shown in Figure 3, with exposure dose ∼0.5 J/mm². The UV laser beam is expanded by a pair of lenses, then split into two orthogonal linearly polarized beams by a polarizing beam-splitter. The two beams are redirected by two mirrors, and pass through quarter-wave (QW) plates (at 45°) to obtain two orthogonal circularly polarized beams that interfere with the BY layers. The two beams should have the same size and power to obtain an ideal pattern. The interference results in a pattern of linearly polarized light, with rotating polarization orientation. For diffraction of the second beam, a second region is illuminated with the same setup, but now with the quarter wave plates at −45°.

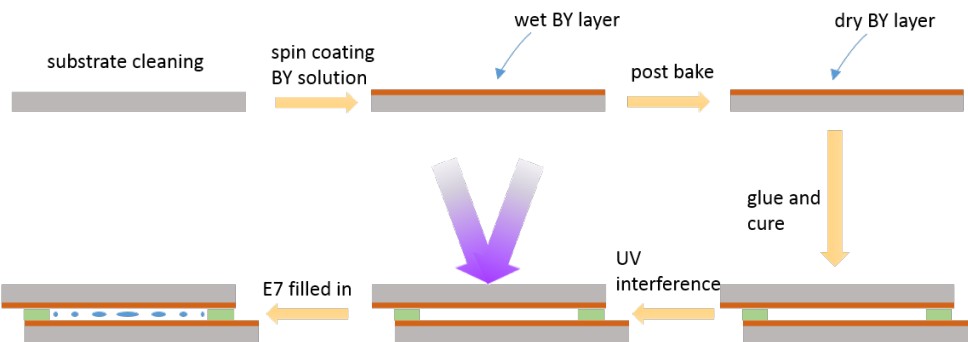

**Figure 2.** Procedure to fabricate a photoaligned LC geometric phase grating.

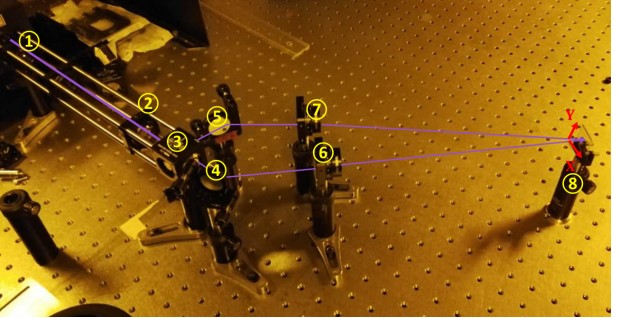

**Figure 3.** Set-up for writing the photoalignment pattern, with (1) (2) lenses to expand the UV beam; (3) polarizing beam splitter; (4) (5) mirrors; (6) (7) quarter-wave plates; (8) empty LC sample.

After the sample is illuminated, the nematic liquid crystal E7 mixture is filled into the cell at a temperature above the clearing point. The cell is filled by capillary action. After cooling, the LC director obtains the desired orientation pattern.

## 3. Results

### 3.1. Characterization of the Gratings

The zero-order transmission of the fabricated device is measured with the same spectrophotometer, and the obtained spectrum is given in Figure 4. As the device only measures light that is not diffracted, this measured transmission should be as close to zero as possible. The lowest transmission is 1.68% and occurs for 647 nm. At the target wavelength of 633 nm, the transmission is still quite low (1.97%).

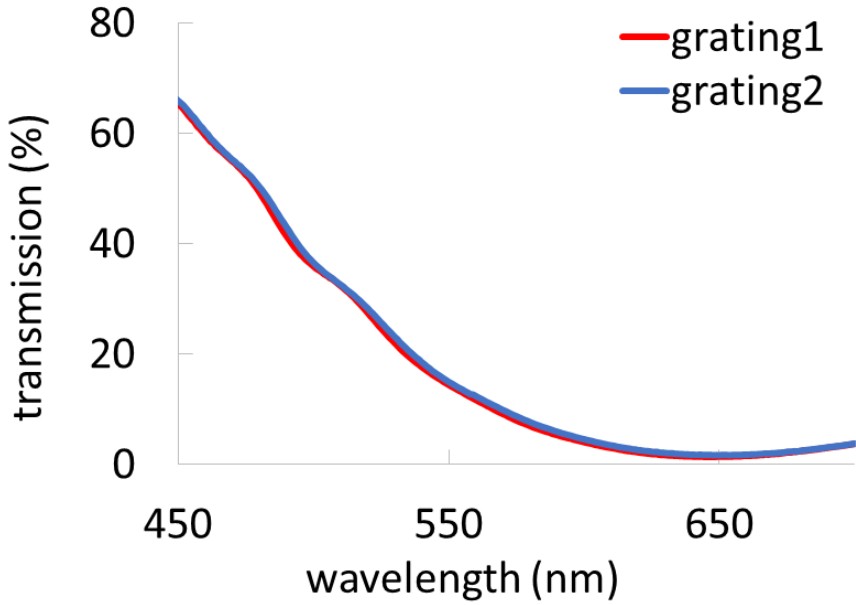

**Figure 4.** Zero-order transmission spectrum for two LC geometric phase gratings.

Figure 5 shows an LC phase grating observed with a polarization microscope. This allows to estimate the period of the grating as 3.4 μm, corresponding to a diffraction angle of 10.7° for 633 nm wavelength. The measured diffraction angle for light of a HeNe laser (633 nm) corresponds to this angle.

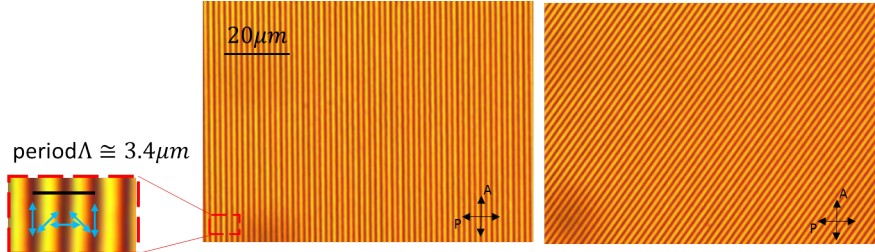

**Figure 5.** Polarization microscopy images of the LC phase grating, with the grating parallel to one of the polarizers (left) or rotated over 45 degrees (right). The inset at the left illustrates the grating period.

### 3.2. Characterization of the Beam Combiner

The set-up to realize a beam combiner based on the two fabricated LC cells, each having two LC geometric phase gratings, is depicted in Figure 6. A HeNe laser with linear polarization, a non-polarizing 50/50 beam splitter, and a mirror are used to generate two

parallel beams with lateral distance of about 10 mm (similar to the typical distance of two laser beams in a laser bank).

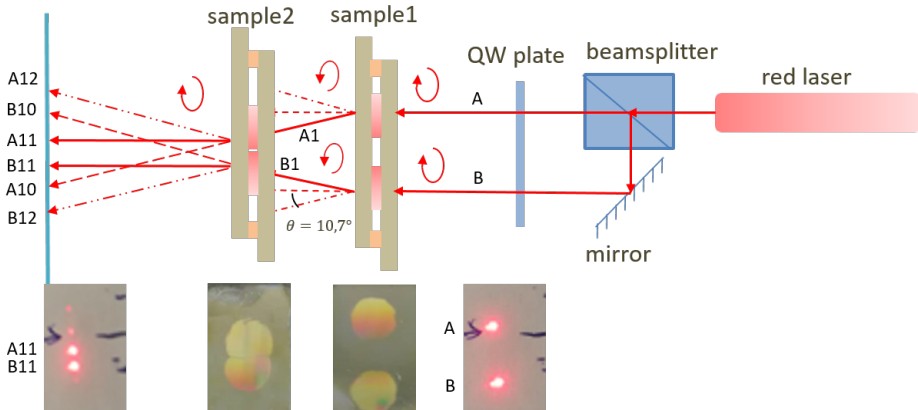

**Figure 6.** Top view of the set-up showing the beam combiner functionality based on two LC phase gratings (samples 1 and 2). A beam splitter is used to make two parallel beams A and B. The combination of samples 1 and 2 brings the parallel beams closer to each other. Six beams arriving on the screen are identified with a code (A10 to B12). The images at the bottom show pictures of the samples, and of the laser spots on the screen, in the absence and presence of the beam combiner.

The two linear beams pass through the same quarter-wave plate with a fast axis rotated over 45°, compared to the polarization direction of the incoming beam, to obtain two beams with circularly polarized light with the same handedness. The two grating patterns on sample 1 have opposite rotation direction, as illustrated in Figure 1. The beams A1 and B1 are the main first-order diffraction orders, and they are circularly polarized with handedness opposite to that of the incoming beams (A and B) and make an angle of 10.7° with the incoming beams. To make the two beams parallel again, a second sample is used. The two grating patterns on the second sample are the same as those on the first sample, but now adjacent to each other. As the handedness of the beams A1 and B1 is opposite to that of beams A and B, the same grating diffracts the beam back into the original direction (solid lines, indicated by A11 and B11). The second sample is placed at a distance of 21 mm from the first sample, to bring the beams 2tan (10.7°) * 21 mm≈ 8 mm closer together: from 10 mm to 2 mm.

Note that the other orders appear on the screen with much lower intensities (dashed lines). The efficiencies for the different diffraction orders for circularly polarized incident light have been measured separately for the four LC phase gratings, and the results are listed in Table 1. The efficiency for a particular diffraction order was determined by dividing the measured power in that order over the total power in all transmitted diffraction orders. The LC phase gratings have glass substrates without anti-reflection coatings, leading to additional losses related to reflection. The diffraction efficiencies for the first orders are all above 98%, which means that the power in the beams A11 and B11 are the most intense. The relative intensities of the different beams encoded in Figure 6 can be estimated by multiplying the corresponding efficiencies for the gratings on the two samples. For example, for beam A12, we expect an intensity proportional to the product of 99.03% (efficiency for sample1-grating1-order1) and 0.03% (efficiency for sample2-grating1-order2).

**Table 1.** The efficiency of the grating.

| Diffraction Order | 2 | 1 | 0 | −1 | Total |
|---|---|---|---|---|---|
| sample1-grating1 | 0.07% | 99.03% | 0.71% | 0.19% | 100% |
| sample1-grating2 | 0.04% | 98.07% | 1.69% | 0.19% | 100% |
| sample2-grating1 | 0.03% | 98.50% | 1.28% | 0.19% | 100% |
| sample2-grating2 | 0.07% | 99.28% | 0.61% | 0.03% | 100% |

### 4. Conclusions

In this work, we demonstrated the realization of a compact, high-efficiency beam combiner, based on photo-patterned LC Pancharatnam–Berry phase gratings. The beam-combining function was realized with four phase gratings with the same period but different rotation directions, and the thickness of LC layer was obtained from optical measurements. The efficiency of the four fabricated gratings for the desired transmission order at 633 nm was above 98%, and the measured diffraction angle (10.7°) matched well with the designed angle. Therefore, we have successfully demonstrated that the lateral distance between the two parallel beams can be efficiently reduced. The principle of this beam combiner could be used to bring more beams closer to each other, for example to reduce the area (and increase the power per unit area) of an array of similar lasers.

**Author Contributions:** Experiments, B.G.; writing—original draft preparation, B.G.; writing—review and editing, K.N. and J.B.; supervision, K.N. and J.B.; project administration and funding acquisition, K.N. and J.B. All authors have read and agreed to the published version of the manuscript.

**Funding:** The work was funded by Flanders Innovation & Entrepreneurship under grant HBC.2017.0141.

**Data Availability Statement:** All the data is contained within this article.

**Conflicts of Interest:** The authors declare no conflict of interest.

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
