# Peer review of "Design and Realization of a Compact Efficient Beam Combiner, Based on Liquid Crystal Pancharatnam–Berry Phase Gratings"

_crystals, doi:10.3390/cryst11020220_

Round 1

Reviewer 1 Report

This work reports on the design and experimental realization of a laser beam combiner based on two parallel diffraction gratings. The gratings are produced by UV photo-patterning which insures periodically rotating alignment of nematic liquid crystal molecules, resulting in a smooth periodic modulation of the geometric phase. An incident circularly polarized beam is strongly deflected into the +1 or -1 diffraction order, depending on the polarization handedness and the rotation direction of the optical axis with respect to the axis of modulation. Each grating has two parts, one for each of the incident parallel beams. The optical axis rotation directions in the two parts are opposite, and chosen so that the beams with the same circular polarization were deflected towards each other by the first grating, and returned to parallel propagation by the second one.

The manuscript is logically structured, well written, and supported by adequate figures and references. I recommend it for publication in Crystals.

Some minor issues which should be amended:
- Eq.1 has different signs for \eta and S_3, but this is true for only one pattern, A or B. For the second pattern, the signs shall be the same. Please specify the pattern and the way to define S_3 (looking along or against the propagation axis, z).
- Numbers in Fig.3 are hard to see, the contrast could be improved.
- Fig.6: the quarterwave plate should not be curved. Also, it would be nice to add images of the beam spots before and after the combiner.

Reviewer 2 Report

In the manuscript, the authors reported on the description of the design of a compact efficient beam combiner, based on liquid crystal Pancharatnam-Berry phase gratings. A nematic liquid crystal mixture E7 was utilized as an anisotropic material. 
The presented manuscript seems to be interesting for a wide scope of journal readers. 
The manuscript is clearly written. I have found only minor issues that should be considered in the revised manuscript:
Figure 5: The scale bars on polarization microscopy images would improve the readability.
Line 98: The temperature of the clearing point of the material should be given.
The operating temperature range of the proposed beam combiner should be estimated.
